# β-Caryophyllene as a Potential Protective Agent Against Myocardial Injury: The Role of Toll-Like Receptors

**DOI:** 10.3390/molecules24101929

**Published:** 2019-05-19

**Authors:** Nancy S. Younis, Maged E. Mohamed

**Affiliations:** 1Department of Pharmaceutical Sciences, College of Clinical Pharmacy, King Faisal University, 31982 Al-Ahsa, Saudi Arabia; MAGED789@HOTMAIL.COM; 2Department of Pharmacology, Zagazig University, Zagazig 44519, Egypt; 3Department of Pharmacognosy, College of Pharmacy, Zagazig University, Zagazig 44519, Egypt

**Keywords:** β-caryophyllene, heat shock protein 60, isoproterenol, MyD88, toll-like receptors

## Abstract

Myocardial infarction (MI) remains one of the major causes of mortality around the world. A possible mechanism involved in myocardial infarction is the engagement of Toll-like receptors (TLRs). This study was intended to discover the prospective cardioprotective actions of β-caryophyllene, a natural sesquiterpene, to ameliorate isoproterenol (ISO)-induced myocardial infarction through HSP-60/TLR/MyD88/NFκB pathway. β-Caryophyllene (100 or 200 mg/kg/day orally) was administered for 21 days then MI was induced via ISO (85 mg/kg, subcutaneous) on 20th and 21st days. The results indicated that ISO induced a significant infarcted area associated with several alterations in the electrocardiogram (ECG) and blood pressure (BP) indices and caused an increase in numerous cardiac indicators such as creatine phosphokinase (CPK), creatine kinase-myocardial bound (CK-MB), lactate dehydrogenase (LDH), and cardiac tropinine T (cTnT). In addition, ISO significantly amplified heat shock protein 60 (HSP-60) and other inflammatory markers, such as TNF-α, IL-Iβ, and NFκB, and affected TLR2 and TLR4 expression and their adaptor proteins; Myeloid differentiation primary response 88 (MYD88), and TIR-domain-containing adapter-inducing interferon-β (TRIF). On the other hand, consumption of β-caryophyllene significantly reversed the infarcted size, ECG and BP alterations, ameliorated the ISO elevation in cardiac indicators; it also notably diminished HSP-60, and subsequently TLR2, TLR4, MYD88, and TRIF expression, with a substantial reduction in inflammatory mediator levels. This study revealed the cardioprotective effect of β-caryophyllene against MI through inhibiting HSP-60/TLR/MyD88/NFκB signaling pathways.

## 1. Introduction

Myocardial infarction (MI), a consequence of ischemic heart diseases, may involve inflammation, which is essential for protection against injury and tissue restoration; however, extreme myocardial inflammation could be a causative agent in further cardiac damage [1]. Therefore, understanding the molecular mechanisms modifying the inflammatory response subsequent to myocardial infarction is critical [2]. Myocardial infarction initiates severe inflammatory responses that intensify irreversible tissue damage [3]. Myocardial necrosis is accompanied with free radical generation; it elicits cytokine cascades, chemokine up-regulation [4], and triggers nuclear factor (NFκB) activation [5]. Necrotic cardiomyocyte generates Damage-Associated Molecular Patterns (DAMPs), which stimulate and trigger intense inflammatory reaction [6,7]. One of the DAMPs are heat shock proteins (HSPs), which are defensive proteins that help cells to be resistant to stress-induced cell destruction [8]. Among the HSPs, HSP-60 which is expressed under stress conditions and serves as a molecular chaperone to enable protein folding and substrate for TLRs [9].

Toll-like Receptor (TLR) pathways prompt NFκB triggering and the up-regulation of chemokine as well as cytokine creation in the infarcted heart [5]. The HSP-60/TLR axis plays a pathogenic role in eliciting cardiomyocyte apoptosis throughout myocardial injury [10]. TLRs role is well recognized in different cardiovascular diseases [11]. After TLR activation, MyD88 is enrolled to the TIR domain (TIR-domain-containing adapter-inducing interferon-β) located in the cytoplasm, where it accelerates IL-1R-associated kinase (IRAKs) association and TNF-receptor associated factor 6 (TRAF6) [12]. TRAF6 provokes transforming growth factor β-activated kinase (TAK) activation, which promotes recruitment of the kinase cascade [11,12], causing translocation NFκB to the nucleus, and facilitating gene transcriptions [13].

Of all the TLRs, TLR4 and TLR2 have been shown to be associated with the development and progression of cardiovascular diseases. TLR4 initiates the expression of a number of pro-inflammatory genes, cell surface molecules, and chemokines, exacerbating myocardium damage [1]. In a rat model of post-infarct heart failure (HF), TLR4 mRNA expression and protein levels were increased in the infarcted and remote myocardium [14]. Additionally, TLR2 was also identified as a death receptor, promoting apoptosis, mediated heart motion abnormalities, inflammation, and fibrosis in MI and HF [14].

β-Caryophyllene is a natural sesquiterpene present in the essential oils of several plants such as cloves, cinnamon, black pepper, and basil [15]. It is a harmless compound with no observed adverse effects [16]; its structure is shown in Figure 1. It is permitted by the United States Food and Drug Administration (FDA) as a food additive, taste enhancer, and natural flavoring compound [17]. β-Caryophyllene (BCP) exerts various documented pharmacological actions including anti-inflammatory [18], cytotoxic [19], antioxidant [20,21], antispasmodic [22], anticancer [21], antimicrobial [21], hypolipidemic [23], and neuroprotective effects [24]. It efficiently diminishes lipopolysaccharide-stimulated TNF-α and IL-1β production [15], and exerts diverse anti-inflammatory and cytoprotective effects [15], which represent an important therapeutic target in several diseases [17].

In spite of the confirmed anti-inflammatory effect of β-caryophyllene, its effect on the cardiovascular system, especially myocardial infarction, has not yet been explored. Accordingly, the existing study was proposed to estimate the cardioprotective effects of β-caryophyllene and to reveal the underlying mechanisms. We tried to identify an alternative mechanism for β-caryophyllene other than that of the cannabinoid type 2 receptors. We also tried to explore endogenous TLR ligands and HSP-60/TLR/MyD88/NFκB signaling pathway to resolve their role in myocardial infarction through NFκB.

## 2. Results

### 2.1. Effects of β-Caryophyllene on Myocardial Infarct Size

Figure 2 illustrates the extent of the infarcted area in different treatment groups. In the normal control and β-caryophyllene rats showed minimal infarcted regions, whereas rats suffer from MI demonstrated significant (*p* < 0.05) 45.16% infarcted regions. β-Caryophyllene (100, 200 mg/kg) treatment showed significant infracted limiting effect (*p* < 0.05) upto 29.82% and 21.68%, respectively (Figure 2). Representative pictures of the TTC-stained rat-heart slices are shown in Figure 2.

### 2.2. Effects of β-Caryophyllene on Electrocardiographic (ECG) Traces

The electrocardiographic traces of normal and experimental animals is shown in Figure 3. In the normal control and β-caryophyllene (BCP) rats showed normal ECG patterns, whereas rats with MI due to ISO showed a significant (*p* < 0.05) increase in the ST segment and QT interval. Conversely, a significant (*p* < 0.05) decrease in P wave, QRS complex, and P-R and R-R intervals were seen in comparison with the control rats. All these diversifications are indicative of infarcted myocardium. Treatment with β-caryophyllene (100, 200 mg/kg) showed a reversal in ECG alterations, causing reduction in both the increase in ST segment and QT interval and the decrease in P wave, QRS complex, and P-R and R-R intervals (Figure 3).

### 2.3. Effects of β-Caryophyllene on Blood Pressure (BP) Indices

ISO induced a significant decrease in heart rate (HR), systolic arterial pressure (SAP), diastolic arterial pressure (DAP), and mean arterial pressure (MAP), measuring 213.3 ± 21.2 beat/min, 76.83 ± 4.1, 73.33 ± 7.25, and 77.6 ± 11 mmHg respectively, as compared to normal control group which measured 375 ± 40.7, 151.17 ± 6.7, 113.17 ± 6.9, and 163.83 ± 8.8 (*p* < 0.05) (Figure 4). However, treatment with β-caryophyllene (100, 200 mg/kg) significantly (*p* < 0.05) prevented the ISO-induced decline in arterial pressure indices. As for SAP, both doses of β-caryophyllene (100, 200 mg/kg) caused an increase, measuring 100.33 ± 4.1 mmHg and 120.83 ± 5.6 mmHg respectively, in relation to the ISO group. Likewise, both DAP and MAP showed a significant (*p* < 0.05) improvement with β-caryophyllene (100, 200 mg/kg) treatment. Similarly, HR was also normalized by BCP (100, 200 mg/kg) treatment as compared to ISO control rats (*p* < 0.05). β-caryophyllene effects on BP and ECG together with the infarct-limiting effect indicated a significant improvement in myocardial functional recovery.

### 2.4. Effects of β-Caryophyllene on Cardiac Marker Enzymes

The effects of isoproterenol (ISO) and β-caryophyllene (100, 200 mg/kg) on numerous cardiac indicator enzymes (lactate dehydrogenase (LDH), creatine phosphokinase (CPK), creatine kinase-myocardial bound (CK-MB), and cardiac tropinin T (cTnT)) in normal and myocardial infarctions induced via isoproterenol are shown in Figure 5. Myocardial infarction subsequent to ISO significantly (*p* < 0.05) augmented cardiac indicator enzymes (LDH, CK-MB, CPK, and cTnT) (Figure 5A–D) as compared with normal rats. Oral treatment with β-caryophyllene (100, 200 mg/kg) significantly (*p* < 0.05) ameliorated the ISO-induced escalation of the diagnostic cardiac indicator enzymes (LDH, CK-MB, CPK, and cTnT), in comparison with the myocardial infarction rats. No significant alteration was detected in the cardiac indicator enzymes in rats treated with only β-caryophyllene.

### 2.5. Effects of β-Caryophyllene Treatment on TLR4, TLR2, MyD88, and TRIF Expression Levels

Real-time quantitative PCR results illustrated that ISO-induced myocardial infarction significantly increased TLR2, TLR4 and their adaptor proteins MYD88, and TRIF expression as shown in Figure 6. Treatment with β-caryophyllene in both doses significantly (*p* < 0.05) decreased TLR2 expression compared to the myocardial infarction group, measuring 4.51 and 3.28 vs. 7.00, respectively, as shown in Figure 5A. Additionally, treatment with β-caryophyllene significantly (*p* < 0.05) diminished TLR4 expression compared to the ischemic group, measuring 4.65 and 3.18 vs. 6.5, respectively (Figure 6B). Furthermore, treatment with β-caryophyllene significantly reduced MYD88 and TRIF relative expression levels compared to the ischemic group (Figure 6C,D).

### 2.6. Effect of β-Caryophyllene Treatment on Toll-Like Receptor Pathway

TLR4, TLR2, MyD88, and TRIF protein levels via western blot in the heart tissue were measured following ISO-induced myocardial infarction to determine whether β-caryophyllene could suppress TLR activity through MyD88 down-regulation. ISO triggered a significant increase in the protein content of TLR4, TLR2, MyD88, and TRIF as compared with normal rats (*p* < 0.05). β-Caryophyllene alone had no effect on the TLR-4, TLR-2, MyD88, and TRIF protein content in the heart tissue in comparison with the normal group. β-Caryophyllene at doses of 100 and 200 mg/kg lessened post- myocardial infarction TLR activity measured as the protein content of TLR2 (A), TLR4 (B), MyD88 (C), and TRIF (D) in the heart tissue (*p* < 0.05) (Figure 7).

### 2.7. Effect of β-Caryophyllene Treatment on HSP-60 and Inflammatory Mediators

HSPs are responsible for a variety of tasks, including immunomodulation, intracellular assembly, folding, and protein translocation, and have been involved as stress proteins in response to myocardial ischemic injuries [25]. The effects of isoproterenol and β-caryophyllene (100, 200 mg/kg) on HSP-60 and inflammatory markers in normal and ISO-induced myocardial infarction are presented in Figure 8. Isoproterenol-induced myocardial infarction significantly amplified HSP-60 and inflammatory markers (TNF-α, IL-1β, and NFκB) as compared with normal rats. No significant alteration was detected in the serum levels of HSP-60 and inflammatory markers in rats treated with only β-caryophyllene. Treatment with β-caryophyllene at a dose of 100 mg/kg decreased HSP-60 and inflammatory marker levels significantly (*p* < 0.05); while β-caryophyllene (200 mg/kg) had a significant effect as regards normalizing HSP-60 and inflammatory marker level.

## 3. Discussion

The administration of high doses of isoproterenol, a β-adrenoceptor agonist, is a commonly used valid experimental approach to induce myocardial infarction [26]. Isoproterenol-induced myocardial infarction in animals, as well as human myocardial infarction, are accompanied by a marked inflammatory response that contributes to additional myocardial injury [27]. Cardiac markers, such as CPK-MB, cTnT, and LDH, are highly sensitive and are specific markers of myocardial cell injury, only being released after myocardial necrosis [28]. In the current study, rats administered with ISO showed a significant infarcted region and an increases in cardiac marker levels with subsequent alterations in heart functions as demonstrated from blood pressure tests and ECG. These findings are in line with previous reports, demonstrating necrotic myocardium damage and leakiness of the plasma membrane due to Isoproterenol-induced myocardial infarction [27,28,29].

Treatment with β-caryophyllene resulted in a lessening of the infarcted region, lowered levels of cardiac markers in the serum, improved ECG, and a reversal in the blood pressure, indicating the protective effects of this treatment and the significant improvement in myocardial functional recovery.

Pursuing inflammatory pathways following myocardial infarction and understanding the immune system role in cardiac restoration is fundamental for optimal approaches for cardiac regeneration. Toll-like receptors are implicated in numerous cardiovascular diseases including myocardial infarction [30,31]. TLR4 and TLR2 were identified as dynamic in mediating the inflammatory response in the ischemic heart [6]. TLR signaling is divided in two main pathways, a MyD88-dependent and a TRIF-dependent pathway [32,33]. MyD88 is a widespread adaptor that motivates the inflammatory pathway [13]. In this pathway, growth factor β-activated kinase 1 (TAK1), a mitogen-activated protein triple kinase (MAPKKK), activates two downstream pathways involving the inhibition of the NFκB kinase (IKK) complex and the MAPK family [12,13]. The IKK complex catalyzes causing the phosphorylation of IκB proteins, leading to NF-κB translocation to the nucleus, subsequently resulting in genes encoding pro-inflammatory cytokines and chemokines transcription [13]. Earlier studies showed that mice with the inactive TLR4 mutant or those genetically deficient in TLR4, TLR2, or MyD88 exhibited reduced MI sizes compared with wild-type, suggesting that TLR2 and TLR4 signaling contributed to ischemic injury in the heart. However, the in vivo studies may have been complicated by the systemic deficiency/inhibition of TLR signaling [34].

Our results showed that ISO-induced myocardial infarction is accompanied by significant escalations in levels of the TLR2, TLR4, MyD88, and TRIF, enhancing subsequent inflammatory mediator signaling. Several previous reports have demonstrated similar increases in TLR2, TLR4, and their adaptor proteins, MyD88 and TRIF, which are induced with ISO (Figure 9) [32,33].

β-Caryophyllene provokes a full agonist action on cannabinoid type 2 (CB2) receptors, which represent an important therapeutic target in several diseases [17]. However, in this study, we tried to find an alternative pathway through which β-caryophyllene might produce cardioprotective effects. β-Caryophyllene treatment causes inhibition of TLR signaling via lowering the up-regulation of TLR2, TLR4, and their adaptor proteins MyD88 and TRIF induced by ISO. Additionally, β-caryophyllene treatment decreased TLR pathway protein expression induced by isoproterenol. These findings are supported by Cho et al. [35] who established that β-caryophyllene protects against d-galactosamine (GalN)/LPS-induced liver injury through diminishing TLR4 and receptor for advanced glycation end product (RAGE) signaling in Kupffer cells. Moreover, Sharma C et al. (2016) proved that β-caryophyllene inhibits pathways triggered by TLR activation, thus reducing immune-inflammatory processes [17]. Several studies showed that targeted disruption or deletion of TLR4 and TLR2 or the adapter protein MyD88 improves cardiac function and reduces the infarct size [30,36]. All of the studies mentioned above suggest that suppression of TLR activation and inhibition of the consequent inflammatory responses are optional goals for the protection from post-myocardial infarction dysfunction and remodeling.

Therefore, this study showed that isoproterenol-induced myocardial infarction caused significant increases in the level of HSP-60 compared to the normal group, while β-caryophyllene treatment significantly diminished this amplification. Studies have shown that ischemia induces obvious HSP-60 release from cardiomyocytes, which can trigger TLR expression leading to induction of cytokine expression [36,37]. TLR4 cardiomyocytes not only have a higher binding capacity for HSP-60, but also mediate robust production of cytokines in response to HSP-60 [36,38]. HSP-60 prompts apoptosis partially via TLR4- and NFκB-dependent mechanisms [25]. TLR4 deletion and HSP-60 blocking markedly reduced compound cytokine gene expression, supporting the concept that HSP-60/TLR4 signaling plays an indispensable part in myocardial inflammation [25].

This study offers perceptions into the core mechanisms by which HSP-60 causes cardiomyocytes inflammation. HSP-60 augmented both TLR2 and TLR4 expression, through MyD88, TRIF, and eventually NFκB (Figure 6). HSP-60/TLR4 mediated cytokine production depended on the activation of MyD88, TRIF, and NF-κB. Thus, the present study provides unique findings concerning the complex interaction between HSP-60 and innate immunity.

The results of the present study revealed a significant reduction in the expression of the inflammation markers TNF-α, IL-Iβ, and NFκB, which mainly evolve in the serum during inflammation. Several studies have shown the effects of isoproterenol on inflammatory release [28]. Treatment with β-caryophyllene, however, resulted in lowered activities of all measured inflammatory mediators, resulting in protection against inflammatory-induced damage in the heart cells. Basha and Sankaranarayanan [18] proved that β-caryophyllene administration significantly decreased pro-inflammatory cytokines in diabetic rats. In addition, β-caryophyllene was found to significantly inhibit the release of pro-inflammatory cytokines, including IL-1β, TNF-α, IL-6, and NFκB, through the activation of BV2 microglia following hypoxic exposure [39].

In conclusion, studies leading to advancements or repositioning concerning drugs, especially those related to natural products in the field of myocardial infarction, should be encouraged, particularly those studies revealing the mechanisms of disease induction and cure. In this study, the author tried to reveal part of the mechanism of myocardial infarction induction by investigating the effect of β-caryophyllene on a model of myocardial infarction. Treatment for 21 days with β-caryophyllene at 100 and 200 mg/kg attenuated cardiac markers and decreased the production of HSP-60, which resulted from isoproterenol-induced myocardial infarction. The cardiac function improvement was associated with the inhibition of HSP-60 and subsequently TLR2, TLR4, and their adaptor proteins MYD88 and TRIF, the production and activity of which ultimately lead to a reduction in inflammatory responses. Finally, treatment with β-caryophyllene reduces the levels of the inflammatory markers TNF-α, IL-1β, and NFκB in heart tissue, which are elevated in isoproterenol-induced myocardial infarction. To the author’s knowledge, this study is the first to establish the cardioprotective effect of β-caryophyllene.

## 4. Materials and Methods

### 4.1. Materials

β-Caryophyllene (BCP) (Cat. No 22075), isoproterenol (ISO) (Cat. No I6504), Creatine Phosphokinase (CPK) ELISA kit (Cat. No. C3755), and triphenyltetrazolium chloride (TTC) (Cat. No 298-96-4)) were obtained from Sigma-Aldrich (St. Louis, MO, USA). Cardiac tropinin T (cTnT) ELISA kit (Cat. No. mbs162871) and Heat Shock Protein 60 (HSP-60) kit (Cat. No. MBS033925) were bought from MyBioSource (San Diego, CA, USA). Creatine Kinase Myocardial Bound (CK-MB) (Cat. No. ABIN955837), Tumor Necrosis Factor-α (TNF- α) (Cat. No. ab46070), Interleukin-1β (IL-1β) (Cat. No. ab100768), and Nuclear Factor-κB (NFκB/p65) (Cat. No. ab133112) ELISA kits were purchased from Abcam Co., Ltd. (Eugene, ORE, USA), Lactate Dehydrogenase (LDL) ELISA kit (Cat. No. E1864r) was bought from Wuhan EIAab Science Co., Ltd. (Wuhan, China). Toll-like receptors TLR4 (SC-10741), TLR2 (SC-10739), MyD88 (SC-11356), reference gene (β-actin; SC-130656), goat anti-rabbit immunoglobulin (Ig) G-horseradish peroxidase (HRP) (SC-2030) antibodies, luminol reagent (SC-2048), and polyvinylidene fluoride (PVDF) membrane (SC-3723) were purchased from Santa Cruz Biotechnology, Inc. (Dallas, TEX, USA). Additional chemicals and reagents used were of the uppermost analytical rank acquired from commercial sources.

### 4.2. Animals

Thirty male Sprague–Dawley rats, weighing 170 ± 25 g, were purchased from the animal house facility, King Saud University, Riyadh, and kept in standard laboratory conditions (23 ± 1 °C) and maintained on a standard commercial rodent diet using a 12 h light/dark cycle during the accommodation period. All animal experimental procedures and protocols were approved by the Animal Research Ethics Committee at King Faisal University (KFU-REC/2018-2-4) and they were performed in accordance with the Guidelines for the Ethical Conduct for Use of Animals in Research, King Faisal University.

### 4.3. Experimental Design

Rats were allocated into five groups after acclimating to the facility: Group I rats were treated with normal saline by oral gavage tubes for 21 days and 0.5 mL of normal saline subcutaneously (SC) on 20th and 21st day, and functioned as normal control. Group II rats were administered β-Caryophyllene (100 mg/kg/day) orally for 21 days and were considered as β-Caryophyllene control as used in previous studies [18,40]. Group III is the isoproterenol control in which rats were given isoproterenol (85 mg/kg, SC) shots with on 20th and 21st day. Finally, rats that were treated with β-Caryophyllene (100 or 200 mg/kg/day) orally for 21 days and isoproterenol (85 mg/kg, SC) on 20th and 21st day were considered as groups VI and V according to the dose of β-Caryophyllene. Isoproterenol dissolved in normal saline was injected subcutaneously (85 mg/kg) at 24-h intervals for 2 days to cause experimental myocardial infarction as mentioned elsewhere [29]

### 4.4. Electrocardiogram (ECG) and Blood Pressure (BP) Recording and Measurement

At the end of the experiments, urethane-anesthetized rats (1.5 g/kg) were placed in a prone position on a board and an ECG was continuously recorded using noninvasive computerized ECG apparatus from Kent Scientific (Torrington, CT, USA). Five minutes later, the ECG was recorded for five seconds. Heart rate, ST segment, P wave, QT, P-R and R-R intervals, and QRS complex were calculated from ECG recordings electronically. While BP measurements were preformed using noninvasive computerized tail-cuff system from Emka Technologies’ Systems (Paris, France), which consists of placing a cuff on the animal’s tail to occlude the blood flow. The pressure was raised and then slowly released. The cuff pressure when the pulse signal reappears is intended as the systolic pressure. The cuff pressure when the pulse signal level recovers its initial level is intended to be the diastolic pressure.

### 4.5. Tissue Handling and Biochemical Estimation

Blood samples were collected, and centrifuged (10 min/4000 rpm) to separate serum which was then stored in a −80 °C deep freezer while waiting to be used for biochemical analysis. Thereafter, the anesthetized animals were sacrificed, and the hearts of different investigational groups were separated, stored in a −80 °C deep freezer for subsequent biochemical parameters determination.

### 4.6. Measurement of Myocardial Infarct Size

As described formerly [41], the frozen heart was cut into 4 or 5 transverse sections, stained with 10% triphenyl tetrazolium chloride (TTC) in phosphate buffer (pH 7.4) for 30 min at room temperature, and then fixed in 10% formaldehyde solution for at least 2 h before it was measured using Image J^®^ programme (Scion Corporation, Maryland, USA). The size of the myocardial infarction (appearing as a pale color) was quantified and calculated as percentage of risk area, which is the total area minus the cavities.

### 4.7. Determination of Cardiac Marker Enzymes and Inflammatory Mediators

ELISA kits, following the manufacturer’s instructions, were used to measure serum cardiac marker levels of CPK, LDH, CK-MB, and cTnT. In addition, heart tissue homogenate levels of HSP-60, TNF-α, IL-Iβ, and NFκB were estimated using ELISA kits according to the manufacturer’s directions using a microplate reader SpectraMax i3x (Molecular Devices, Silicon Valley, CA, USA).

### 4.8. Quantitative Analysis of TLR Pathway

Table 1 lists all the primers sequences used for real-time PCR [35,42]. Real-time PCR was done according to the method described elsewhere in [43]. Briefly, Trizol reagent kit (Invitrogen, Carlsbad, CA, USA) was used to extract and purify total RNA from samples followed by exposing to reverse transcription polymerase chain reaction (RT-PCR) kit (TaKaRa, Kusatsu, Shiga, Japan) to reverse transcription reaction. The 20 μL serves as a reaction volume incorporated 1 μL of the total RNA (1 μg/µL), and the reaction was accomplished by incubating at 42 °C for 15 min then at 95 °C for 2 min. The cDNA was stored prior to additional use at −20 °C. The SYBR Green real-time PCR was accomplished with the SYBR ExScript RT-PCR kit (TaKaRa, Kusatsu, Shiga, Japan). Reaction mixture 50 μL enclosed 1 μL of × 50 ROX Reference Dye, 2 μL of sense and antisense primers (1 μL per each primer mentioned in Table 1), 25 μL of × 2 SYBR Green PCR Master Mix, 4 μL of cDNA template, and lastly 18 μL of sterilized distilled H_2_O (dH_2_O). The PCR reaction condition incorporated pre-denaturing at 95 °C for 10 s, then 40 cycles of 95 °C/ 5 s, and 60 °C/30 s and 72 °C/1 min. Quantification analyses were completed via Opticon-2 Real-time PCR reactor (MJ Research, Reno, NV, USA). Step PE Applied Biosystems (Perkin Elmer, Waltham, MA, USA) software was used to analyze real time-PCR results. Expression of the target gene was measured and correlated to the reference gene (β-actin).

### 4.9. Detection of the Toll-Like Receptors Pathway Protein Expressions

Western blot was preformed according to the method described previously [42]. Briefly, heart tissues samples were homogenized in radioimmunoprecipitation assay (RIPA) buffer containing protease inhibitor; the total protein extracted was calculated using a NanoDrop Lite spectrophotometer (Thermo Fisher Scientific, Waltham, MA, USA). Thereafter, 50 μg of the total extracted protein was separated via sodium dodecyl sulfate (SDS)-polyacrylamide gel electrophoresis (PAGE) and blotted onto PVDF membranes. Blocking PVDF membranes were done by incubation in Tris-buffered saline (TBS) enclosing 3% bovine serum albumin and 0.1% Tween 20 for 1 h at room temperature. After washing with TBS containing 0.1% Tween 20, the membranes were incubated firstly with the primary antibodies (1:300 dilution) for 2 h, and then goat anti-rabbit HRP-conjugated (as secondary antibody; at a 1:5.000 dilution) at room temperature. The chemiluminescence produced from the luminol reagent was detected with the C-DiGit chemiluminescence scanner (LI-COR, Lincoln, NE, USA), and the band intensity was analyzed using the scanner software.

### 4.10. Statistical Analysis

All the values were expressed as mean ± SEM (*n* = 6). For western blotting analysis, densitometry analysis was completed using Image J software. GraphPad Prism 5 software (GraphPad Software Inc., San Diego, CA, USA,) was preformed to evaluate the statistical analysis. The value *p* < 0.05 was considered statistically significant using one way analysis of variance (ANOVA) followed by Tukey’s test.

## Figures and Tables

**Figure 1 molecules-24-01929-f001:**
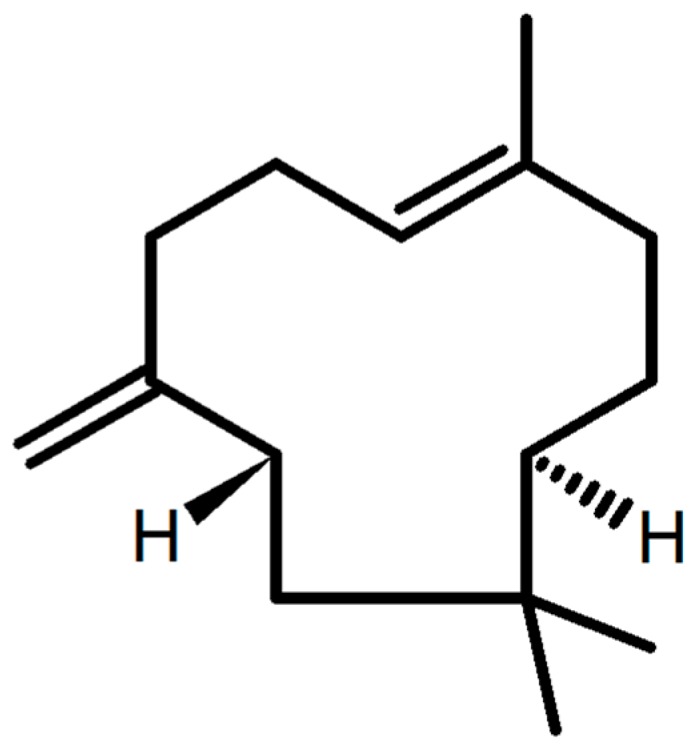
β-Caryophyllene (BCP) structure.

**Figure 2 molecules-24-01929-f002:**
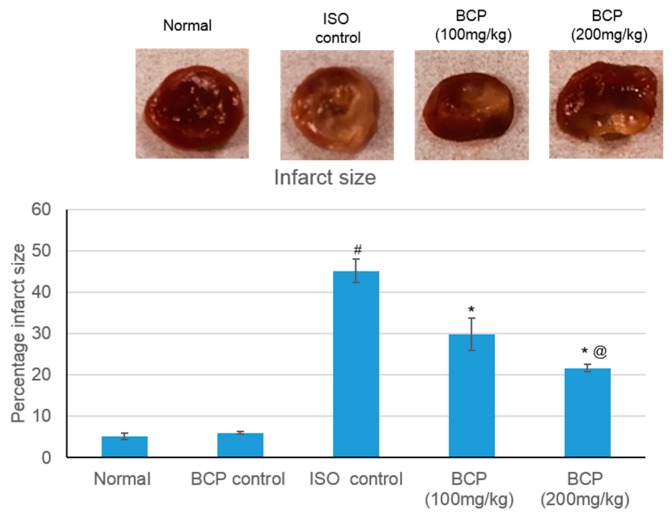
Representative pictures of the TTC-stained rat-heart slices showing the effects of β-caryophyllene treatment (100, 200 mg/kg) for 21 days on myocardial infarction size in ISO-induced myocardial infarction. All values were expressed as mean ± SD (*n* = 6). ISO: isoproterenol; BCP: β-Caryophyllene; TTC: triphenyl tetrazolium chloride. # indicates statistically significant from normal group, * indicates statistically significant from isoproterenol group, @ indicates statistically significant from β-caryophyllene (100 mg/kg) group (*p* < 0.05) using one way ANOVA followed by Tukey’s test as a post-hoc analysis.

**Figure 3 molecules-24-01929-f003:**
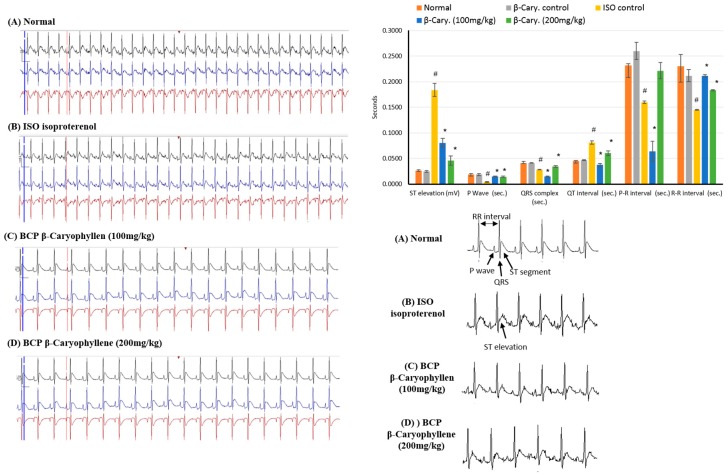
Representative traces of the electrocardiographic (ECG) showing the effects of β-caryophyllene treatment (100, 200 mg/kg) for 21 days on ECG components in ISO-induced myocardial infarction. All values were expressed as mean  ±  SD (*n* = 6). ISO: isoproterenol and BCP: β-Caryophyllene. # indicates statistically significant from normal group, * indicates statistically significant from isoproterenol group, @ indicates statistically significant from β-caryophyllene (100 mg/kg) group (*p* < 0.05) using one way ANOVA followed by Tukey’s test as a post-hoc analysis.

**Figure 4 molecules-24-01929-f004:**
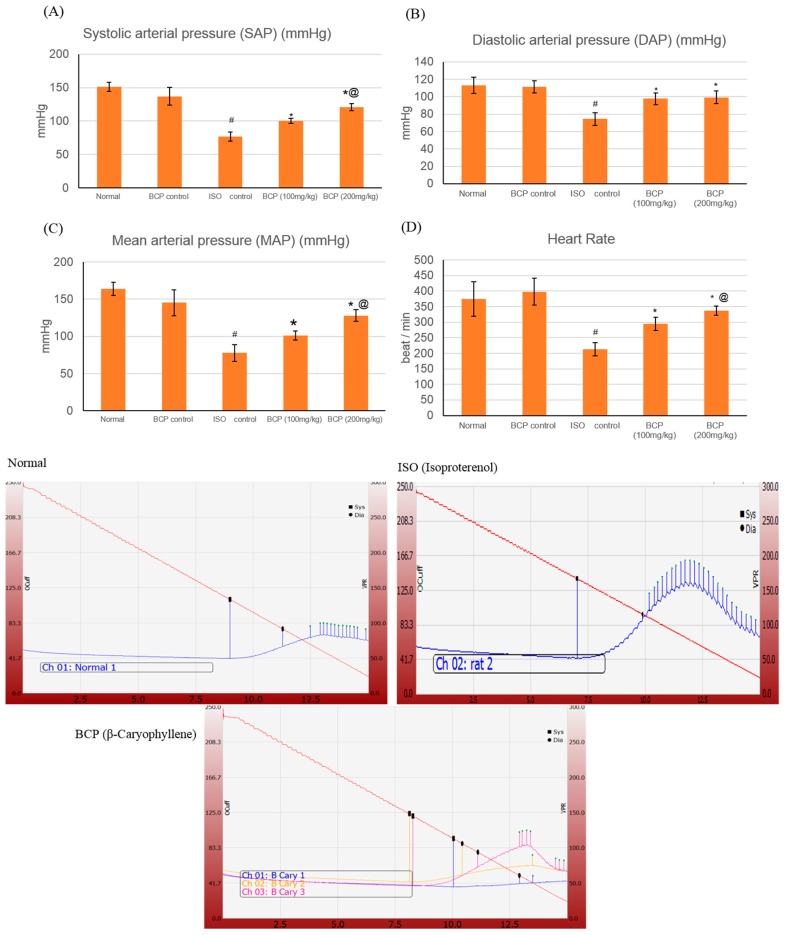
Upper section: effects of β-caryophyllene treatment (100, 200 mg/kg) for 21 days on Blood Pressure (BP) indices in ISO-induced myocardial infarction: (**A**) SAP; (**B**) DAP; (**C**) MAP; (**D**) HR. Lower section: shows representative computerized traces of the β-caryophyllene effects on BP. All values were expressed as mean ± SD (*n* = 6). ISO: isoproterenol; BCP: β-Caryophyllene, SAP: systolic arterial pressure; DAP: diastolic arterial pressure; MAP: mean arterial pressure; and HR: heart rate. # indicates statistically significant from normal group, * indicates statistically significant from ISO group, @ indicates statistically significant from BCP (100 mg/kg) group (*p* < 0.05) using one-way ANOVA followed by Tukey’s test as a post hoc analysis.

**Figure 5 molecules-24-01929-f005:**
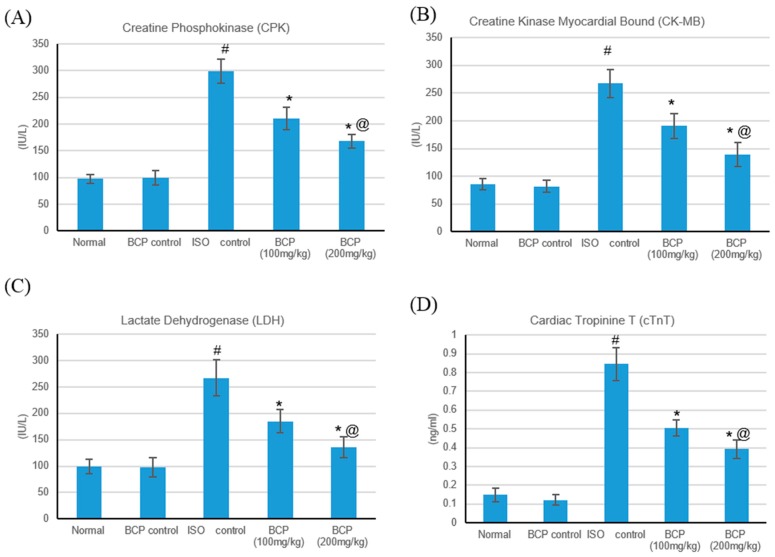
Effects of β-caryophyllene treatment (100, 200 mg/kg) for 21 days on cardiac marker enzymes in ISO-induced myocardial infarction: (**A**) CPK, (**B**) CK-MB, (**C**) LDH, and (**D**) cTnT. All values were expressed as mean ± SD (*n* = 6). ISO: isoproterenol; BCP: β-Caryophyllene; CPK: Creatine Phosphokinase; CK-MB: Creatine Kinase-Myocardial Bound; LDH: Lactate dehydrogenase and cTnT: Cardiac Troponin T. # indicates statistically significant from normal group, * indicates statistically significant from isoproterenol group, @ indicates statistically significant from β-caryophyllene (100 mg/kg) group (*p* < 0.05) using one-way ANOVA followed by Tukey’s test as a post-hoc analysis.

**Figure 6 molecules-24-01929-f006:**
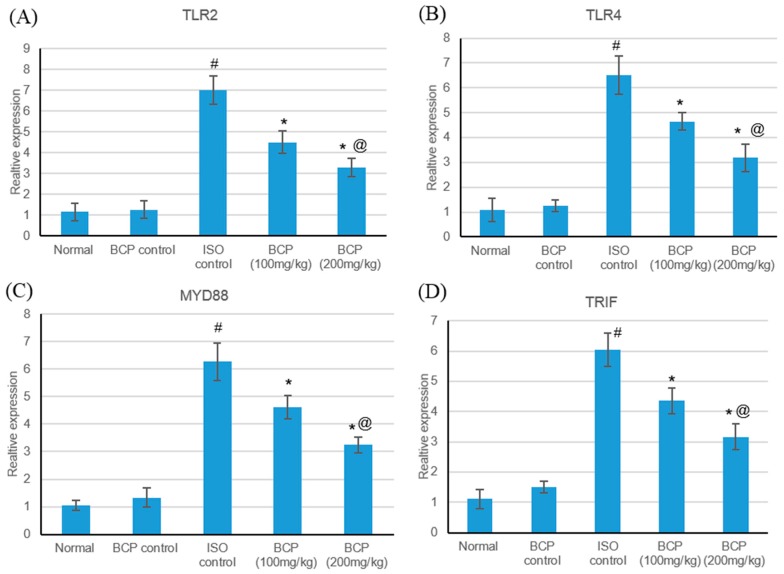
Effects of β-caryophyllene treatment (100, 200 mg/kg) for 21 days on the TLR pathway in ISO-induced myocardial infarction; (**A**) TLR2 mRNA, (**B**) TLR4 mRNA, (**C**) MyD88 mRNA, and (**D**) TRIF mRNA expression levels. All values were expressed as mean ± SD. ISO: isoproterenol; BCP: β-Caryophyllene and TLR: Toll-like Receptors. # indicates statistically significant from normal group, * indicates statistically significant from isoproterenol group, @ indicates statistically significant from β-caryophyllene (100 mg/kg) group (*p* < 0.05) using one-way ANOVA followed by Tukey’s test as a post-hoc analysis.

**Figure 7 molecules-24-01929-f007:**
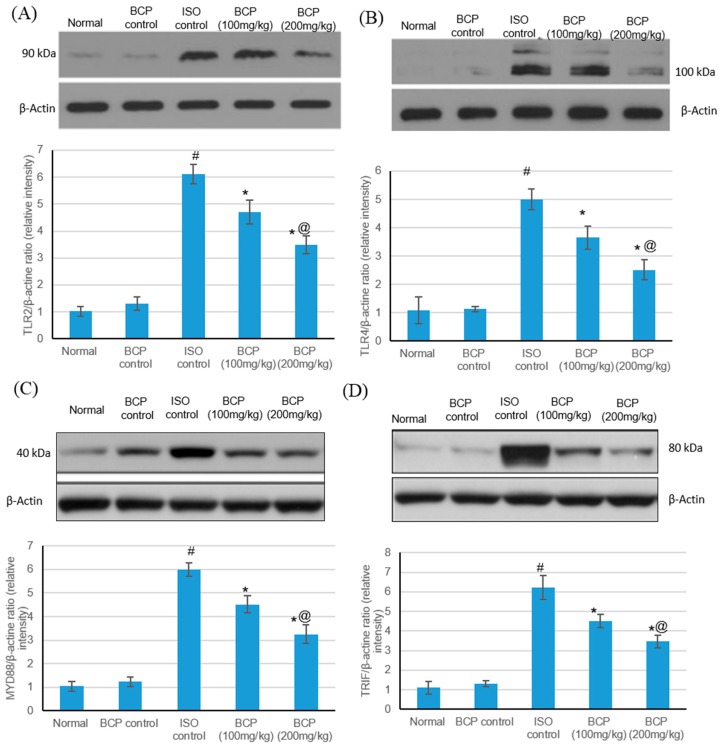
Effects of β-caryophyllene treatment (100, 200 mg/kg) for 21 days on TLR pathway protein content in ISO-induced myocardial infarction: (**A**) TLR2, (**B**) TLR4, (**C**) MyD88, and (**D**) TRIF protein content. Upper section: representative immunoblots of TLR4, TLR2, MyD88, and TRIF protein content in the normal control heart tissues and in the hearts after isoproterenol-induced myocardial infarction in the absence or presence of pre-treatment with the graded doses of β-caryophyllene. **Lower section**: Densitometric analysis of TLR4, TLR2, MyD88, and TRIF. Values are mean ± S.E.M (*n* = 6) for the ratio of TLR4, TLR2, MyD88, and TRIF to β-actin. ISO: isoproterenol and BCP: β-Caryophyllene; # indicates statistically significant from normal group, * indicates statistically significant from isoproterenol group, @ indicates statistically significant from β-caryophyllene (100 mg/kg) group (*p* < 0.05) using one-way ANOVA followed by Tukey’s test as a *post-hoc* analysis.

**Figure 8 molecules-24-01929-f008:**
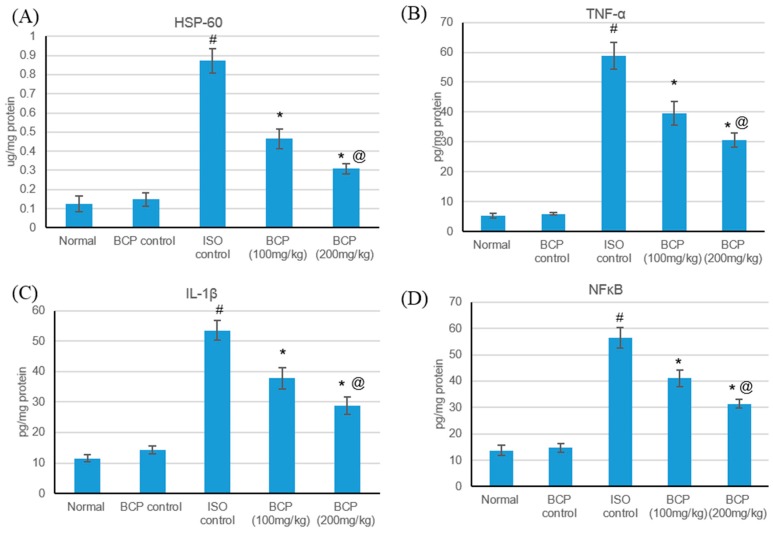
Effects of β-caryophyllene treatment (100, 200 mg/kg) for 21 days on HSP-60 and inflammatory markers in ISO-induced myocardial infarction: (**A**) HSP-60, (**B**) TNF-α, (**C**) IL-1β, and (**D**) NF- κB. All values were expressed as mean ± SD (*n* = 6). ISO: isoproterenol; BCP: β-Caryophyllene; HSP-60: Heat shock protein 60; TNF-α: Tumor necrosis factor–alpha; IL-Iβ: Interleukin-1β and NFκB: nuclear factor-κB. # indicates statistically significant from normal group, * indicates statistically significant from isoproterenol group, @ indicates statistically significant from the β-caryophyllene (100 mg/kg) group (*p* < 0.05) using one-way ANOVA followed by Tukey’s test as a post-hoc analysis.

**Figure 9 molecules-24-01929-f009:**
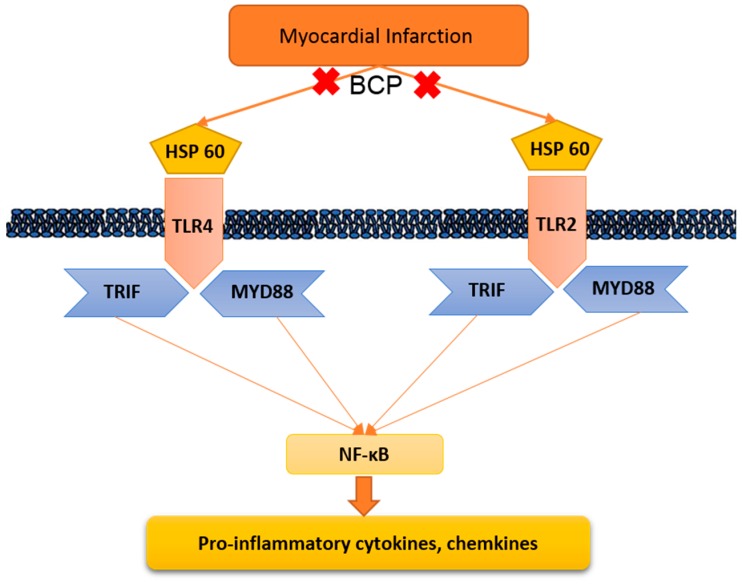
Suggested roll of β-caryophyllene on Toll-like receptor signaling. HSP-60: Heat shock protein 60; TLR2: Toll-like receptor 2; MYD88: Myeloid differentiation primary response gene 88; TRIF: TIR-domain-containing adapter-inducing interferon-β; and NFκB: Nuclear Factor-κB.

**Table 1 molecules-24-01929-t001:** Primers sequence used for real-time PCR.

	Primer Sequence (5′ to 3′)
TLR4	F AGTGTATCGGTGGTCAGTGTGCT R AAACTCCAGCCACACATTCC
TLR 2	F AAACTGTGTTCGTGCTTTCTGA R CTTTCTTCTCAATGGGTTCCAG
MyD88	F GAGATCCGCGAGTTTGAGAC R CTGTTTCTGCTGGTTGCGTA
TRIF	F TCAGCCATTCTCCGTCCTCTTC R GGTCAGCAGAAGGATAAGGAA
β-Actin	F CACGATGGAGGGGCCGGACTCATC R TAAAGACCTCTATGCCAACACAGT

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
