# Peer review of "β-Caryophyllene as a Potential Protective Agent Against Myocardial Injury: The Role of Toll-Like Receptors"

_molecules, 2019, doi:10.3390/molecules24101929_

Round 1

Reviewer 1 Report

Nancy et al., in their manuscript try to show that β-caryophyllene as a potential protective agent against myocardial injury: through the role of Toll-Like Receptors:

Major concerns:

To measure if the model of myocardial infarction by isoproterenol was successful, measurements of infarct size (by triphenyltetrazolium chloride staining), fibrosis (Masson’s trichrome staining), heart functions should be included in the manuscript.

The authors demonstrated how β-Caryophyllene regulated TLRs and reduced inflammatory responses, however which type of cells (e.g. macrophages, cardiomyocytes, neutrophils) did β-Caryophyllene affect particularly has not been shown, more precise in vitro experiments are suggested.

The qPCR and WB results convinced the readers that β-Caryophyllene regulate the expression of TLR2 and TLR4, whereas whether β-Caryophyllene regulates other TLRs is questionable, an RNA-seq or qPCR of more TLRs should be included.

β-Caryophyllene did recover part of cardiac injury which was measured by cardiac marker enzymes, in the meantime, β-Caryophyllene did downregulate the expression of TLR2 and TLR4, whether the cardiac injury was recovered because of the decrease of TLRs’ levels, however, has not been proved. At least an in vitro silencing of TLR2 and TLR4 should be performed to demonstrate whether β-Caryophyllene recovers cardiac injury by downregulating TLR2 and TLR4, instead of by other means.

Minor concerns:

Line 310: LDL should be LDH

Figure5: The names of proteins (e.g. TLR2, ß-actin) should be marked beside the blotting.

Author Response

Reviewer 1 point by point response

Major concerns

Reviewer’s comment:

To measure if the model of myocardial infarction by isoproterenol was successful, measurements of infarct size (by triphenyltetrazolium chloride staining), fibrosis (Masson’s trichrome staining), heart functions should be included in the manuscript

Authors’ response:

Thank you for the comment. we would like to mention that myocardial infarction model using isoproterenol the is a very common, valid and well established model (1). Here; we evaluated the successfulness of the model using the heart function testes (EEG and BP) and cardiac biomarkers (LDH, CK-MB, CPK and cTnT) in addition to infract size suggested by the reviewer. Cardiac troponin T (cTnT) is sensitive and specific markers in detecting myocardial necrosis, and have become the preferred biomarkers for the diagnosis of acute myocardial infraction and is considered the gold-standard in the new definition of myocardial infarction (2). The detection of cTnT in the blood stream is, therefore, a highly specific marker for cardiac damage (3, 4). However; due to the concerns raised by the reviewer; We preformed the measurements of infarct size via triphenyltetrazolium chloride TTC staining using frozen samples stored from all the experimental groups and the results of the new parameter was added in the revised manuscript.  Please refer to section 2.1 in results section and 4.6 in the materials and methods section.

Reviewer’s comment:

The authors demonstrated how β-Caryophyllene regulated TLRs and reduced inflammatory responses, however which type of cells (e.g. macrophages,   cardiomyocytes, neutrophils) did β-Caryophyllene affect particularly has not been shown, more precise in vitro experiments are suggested

Authors’ response:

Thank you for the comment. Yes, we agree with the reviewer’s comment that the type of cells in which the TLRs are regulated by β-Caryophyllene is important, however we think this suggested advanced experiments should be addressed as a further study and not belong to this manuscript. We are just introducing the potential of using β-Caryophyllene as an anti-myocardial infarction agent in this manuscript. If this manuscript should be published, this will open the door for more in-depth experiments to support the idea of β-Caryophyllene as an anti-myocardial infarction agent. And those in-vitro experiments suggested by the reviewer could be a part of this in-depth study.

Another point to consider is that earlier studies have demonstrated that TLRs are expressed in heterogeneous tissues and cells (5, 6). For example, it was proven that TLRs are expressed in both compartments; leukocytes, endothelial cells and cardiomyocytes express TLRs, which upon ligand binding (either PAMPs or DAMPs), cardiomyocytes and endothelial cells undergo the same TLR signal transduction compared to leukocytes (5).

Necrotic cardiac myocytes release a wide range of endogenous DAMPs, cause significant TLR4 induction. TLR4 and downstream gene expression profiles are upregulated in both infarcted and remote myocardium following MI (6). TLR4 is expressed in nonprofessional immunocyte cell types: cardiac myocytes and microvascular endothelial cells (7). Also, Kolodzinska A et al, (2018) found that TLR2, TLR4 protein expression both in cardiomyocytes and in infiltrating mononuclear cells differs among rats which may indicate an individual innate immunity ability to respond to an identical inducible factor (ISO). TLR2, TLR4 may be expressed on the cell surface and in the cytoplasm of the cell (8). Finally, we came to a conclusion that since TLR4 and TLR2 are expressed on both cardiac myocytes and circulating cells, it is not clear which TLR4/2-expressing cells contribute mainly to cardiac dysfunction associated myocardial infraction and this could be a valid point for further study, however as we mentioned above this is not the main scope of this manuscript.

Reviewer’s comment:

The qPCR and WB results convinced the readers that β-Caryophyllene regulate the expression of TLR2 and TLR4, whereas whether β-Caryophyllene regulates other TLRs is questionable, an RNA-seq or qPCR of more TLRs should be included

Authors’ response:

 Thank you for the valuable comment.  In this study we focused on the effect on β-Caryophyllene on TLR2 and TLR4 due to their documented role in Myocardial infraction. TLR are of high number thus the significant of knowing the effect of β-Caryophyllene on each one of these is questionable that is why from all TLRs we choose to investigate only TLR2 and TLR4. TLR2 and TLR4 have been shown to have the most fundamental role in promoting cytokine production and subsequent inflammatory damages in cardiovascular events. A new paragraph was added to introduction (paragraph no 3) explaining the role of both TLR2 and TLR4 in myocardial infraction. Moreover; the effect of β-Caryophyllene on other types of TLRs should be our target in the follwoing study.

Of all the TLRs, TLR4 has been shown to be associated with the development and progression of CVDs. TLR4 initiates the expression of a number of pro-inflammatory genes, cell surface molecules, and chemokines which exacerbates the damage to myocardium (6).

TLR2 role: TLR2 was also identified as a death receptor promoting apoptosis, mediated heart motion abnormalities, inflammation and fibrosis in MI and HF (8).

TLR 4 role: In a rat model of post-infarct HF, TLR4 mRNA expression and protein levels were increased in infarcted and remote myocardium (8).

Reviewer’s comment:

β-Caryophyllene did recover part of cardiac injury which was measured by cardiac marker enzymes, in the meantime, β-Caryophyllene did downregulate the expression of TLR2 and TLR4, whether the cardiac injury was recovered because of the decrease of TLRs’ levels, however, has not been proved. At least an in vitro silencing of TLR2 and TLR4 should be performed to demonstrate whether β-Caryophyllene recovers cardiac injury by downregulating TLR2 and TLR4, instead of by other means

Authors’ response:

The reviewer raised a valid and good point and we thank him for this valuable comment which will indeed enrich the manuscript. We totally agree with the reviewer in that there is a lost ring here, which is the effect of TLRs on the state of myocardial infarction. Many studies have emphasized the relation between TLR 2 and 4 downregulation and the improvement of the cardiac infarction state and to fulfill the reviewer’s concerns, paragraph number three in the discussion section was rephrased to include this particular point. Although we agree with the reviewer’s point, we think that using in vitro silencing approach for TLR 2 and 4 to prove this point is not practical. Firstly; in-vitro silencing techniques do not prevent the gene expression by 100% so the results will not be conclusive or reproducible. We prefer using the knockout techniques, and we think that using all these techniques are not in the scope of this manuscript. Furthermore, we only suggest down regulation of TLRs 2 and 4 as a mechanism for the myocardial infarction recovering activity of β-Caryophyllene.

Secondly, studies from several laboratories using knockout mouse models suggest that TLR2, TLR4, and MyD88 may all contribute to myocardial inflammation and infarction. It is noteworthy, however, that in these models, TLR and MyD88 deficiency are systemic and, therefore, the exact contribution of TLR signaling of cardiac (vs. circulatory) origin is unknown. Nevertheless, the disparate findings illustrate the complexity and difficulties in defining the emerging role of TLRs as a critical modulator in both tissue inflammation and injury. Defining the role of innate immune signaling in ischemic myocardial injury may have important therapeutic implications. 

Previous study done by Yang Y et al, (2016) demonstrated that direct inhibition of TLR4 may not be a promising therapeutic avenue to improve the prognosis of myocardial inflammation to some extent, since TLR4 inhibition may lead to a functional loss of the innate immune mechanism. Thus, although excessive pro-inflammatory cytokine production exerts detrimental effects in myocardial inflammation, some cytokines also drive the host defense and tissue repair process within the heart (6). In addition; Chao W (2009) mentioned that mice with the inactive TLR4 mutant or genetically deficient for TLR4, TLR2, or MyD88 exhibited reduced MI sizes compared with wild type suggesting that TLR2 and TLR4 signaling contributed to ischemic injury in the heart. However, the in vivo studies may have been complicated by the systemic deficiency/inhibition of TLR signaling. This is due that that systemic TLR deficiency also leads to a significant reduction in the level of myocardial inflammation as measured by neutrophil recruitment, an NF-κB-dependent expression of cytokines and chemokines, and a complement deposition in the heart after I/R (9)

Minor concerns:

Reviewer’s comment:

Line 310: LDL should be LDH: corrected in the revised manuscript

Figure5: The names of proteins (e.g. TLR2, ß-actin) should be marked beside the blotting: done in the revised manuscript.

References:

1. Shukla SK, Sharma SB, Singh UR. beta-Adrenoreceptor Agonist Isoproterenol Alters Oxidative Status, Inflammatory Signaling, Injury Markers and Apoptotic Cell Death in Myocardium of Rats. Indian journal of clinical biochemistry : IJCB. 2015;30(1):27-34.

2. Loria V, Leo M, Biasillo G, Dato I, Biasucci LM. Biomarkers in Acute Coronary Syndrome. Biomarker insights. 2008;3:453-68.

3. Garg P, Morris P, Fazlanie AL, Vijayan S, Dancso B, Dastidar AG, et al. Cardiac biomarkers of acute coronary syndrome: from history to high-sensitivity cardiac troponin. Internal and emergency medicine. 2017;12(2):147-55.

4. Park KC, Gaze DC, Collinson PO, Marber MS. Cardiac troponins: from myocardial infarction to chronic disease. Cardiovascular research. 2017;113(14):1708-18.

5. Arslan F, Keogh B, McGuirk P, Parker AE. TLR2 and TLR4 in ischemia reperfusion injury. Mediators of inflammation. 2010;2010:704202.

6. Yang Y, Lv J, Jiang S, Ma Z, Wang D, Hu W, et al. The emerging role of Toll-like receptor 4 in myocardial inflammation. Cell Death Dis. 2016;7:e2234.

7. Frantz S, Kobzik L, Kim YD, Fukazawa R, Medzhitov R, Lee RT, et al. Toll4 (TLR4) expression in cardiac myocytes in normal and failing myocardium. The Journal of clinical investigation. 1999;104(3):271-80.

8. Kolodzinska A, Czarzasta K, Szczepankiewicz B, Glowczynska R, Fojt A, Ilczuk T, et al. Toll-like receptor expression and apoptosis morphological patterns in female rat hearts with takotsubo syndrome induced by isoprenaline. Life sciences. 2018;199:112-21.

9. Chao W. Toll-like receptor signaling: a critical modulator of cell survival and ischemic injury in the heart. American journal of physiology Heart and circulatory physiology. 2009;296(1):H1-H12.

Reviewer 2 Report

The manuscript, molecules-382094, describes that β-Caryophyllene is powerful protective compound against isoproterenol-induced myocardial infarction through Toll-like receptors pathway. It is interesting that the present compound prevents myocardial damage caused by isoproterenol through TLRs’ pathway. However, paper in a current version is needed to correct. It is beneficial to get an English editing service. Some grammar errors were found. Certain statements are difficult to understand. The following major mistakes are noticed:

The way of authors showed their data was inappropriate. When we want to present myocardial infarct-related data, it is better to report some results about clinical features (eg; ECG, BP) as the first data.

In the abstract section, the words count is more than requirement. What do they mean about “laboratory rat”? 

The way how the authors wrote the introduction section doesn’t make sense. They should make this part more interesting.

The authors should provide the structure of β-Caryophyllene.

In the results section, it is found some simple description about all the data. The authors can explain each of their finding rigorously. P value were missing. In the figures, the bar chart were better be showed in black and white colors. The significant symbols should use some common sign. It is supposed that the figures’ legend can describe the figure clearly. In the Figure 2, the EGC figure, the authors should reduce the wave numbers. Thus, it can be clearly seen every heart specific waves (ST elevation, P wave, QRS wave, etc). for ST elevation, they should put the mark on it. There are no (A), (B) part in this figure. In the Figure 3, BP figures are too small. In the figure 5, there no explanation about protein’s name and molecular weight of each protein.

In the 2.3 section, the authors should describe about the blood pressure and heart rate findings.

In the discussion section, the authors should explain their findings more rigorous.

Some typos are found in the manuscript, for example tropinine, infract, etc. Otherwise, it is important to notice about the righteous way of writing.

In line 105, the colon symbol (:) should be removed.

In line 216, the “&” should be changed.

Overall, the authors should revise their manuscript and check it carefully.

Author Response

Reviewer 2 point by point response

Reviewer’s comment: paper in a current version is needed to correct. It is beneficial to get an English editing service. Some grammar errors were found. Certain statements are difficult to understand.

Authors’ response:  English editing was done through the MDI English editing service. Thank you for the suggestion.

Reviewer’s comment: The way of authors showed their data was inappropriate. When we want to present myocardial infarct-related data, it is better to report some results about clinical features (eg; ECG, BP) as the first data

Authors’ response:  Thank you for your comment and was implemented. Please refer to the revised manuscript.

Reviewer’s comment: In the abstract section, the words count is more than requirement. What do they mean about “laboratory rat”?

Authors’ response: corrected as suggested, Thank you for your comment. Please refer to the revised manuscript.

Reviewer’s comment: The way how the authors wrote the introduction section doesn’t make sense. They should make this part more interesting.

Authors’ response:  Thank you for your comment and we made some changes to make the introduction more interesting as suggested besides English editing of the whole manuscript was done as suggested above.

Reviewer’s comment: The authors should provide the structure of β-Caryophyllene

Authors’ response: structure of β-Caryophyllene was mentioned in the introduction section of the revised manuscript as Figure 1.

Reviewer’s comment: 

In the results section, it is found some simple description about all the data. The authors can explain each of their finding rigorously. 

P value were missing.

In the figures, the bar chart were better be showed in black and white colors.

The significant symbols should use some common sign.

It is supposed that the figures’ legend can describe the figure clearly.

In the Figure 2, the EGC figure, the authors should reduce the wave numbers. Thus, it can be clearly seen every heart specific waves (ST elevation, P wave, QRS wave, etc). for ST elevation, they should put the mark on it.

There are no (A), (B) part in this figure.

In the Figure 3, BP figures are too small.

In the figure 5, there no explanation about protein’s name and molecular weight of each protein

Authors’ response: Thank you for all your comments. Please refer to the revised manuscript.

We made some changes to explain each of our finding rigorously together with English editing. 

P value were added.

Online version of the manuscript gives us the chance to use colored bar chart.

We tried to use more common sign for significant symbols.

Figures legends were changes to describe the figure clearly and stand alone.

we reduce the wave numbers and mark ECG component on it and ST elevation was mentioned. 

In the Figure 3, BP figures were changed to be more clear.

In the figure 5 protein’s name and molecular weight of each protein were added.

Reviewer’s comment:  In the 2.3 section, the authors should describe about the blood pressure and heart rate findings

Authors’ response: blood pressure and heart rate findings were added in the 2.3 section as suggested. Please refer to the revised manuscript.

Reviewer’s comment: In the discussion section, the authors should explain their findings more rigorous

Authors’ response: we made some adjustments in the discussion section as suggested.  Thank you

Reviewer’s comment:  Some typos are found in the manuscript, for example tropinine, infract, etc. Otherwise, it is important to notice about the righteous way of writing

Authors’ response: corrected as suggested and English editing was performed.

Reviewer’s comment:  In line 105, the colon symbol (:) should be removed

Authors’ response: corrected as suggested

Reviewer’s comment In line 216, the “&” should be changed

Authors’ response: corrected as suggested

Round 2

Reviewer 1 Report

Some some of my concerns still remains.

Reviewer 2 Report

No comment.